# Pulsed Dipolar EPR for Self-Limited Complexes of Oligonucleotides Studies

**DOI:** 10.3390/biom14080887

**Published:** 2024-07-23

**Authors:** Alexey S. Chubarov, Burkhard Endeward, Maria A. Kanarskaya, Yuliya F. Polienko, Thomas F. Prisner, Alexander A. Lomzov

**Affiliations:** 1Institute of Chemical Biology and Fundamental Medicine SB RAS, 630090 Novosibirsk, Russia; makanarskaya@gmail.com; 2Institute of Physical and Theoretical Chemistry, Goethe University Frankfurt, Max-von-Laue-Str. 7, 60438 Frankfurt am Main, Germany; b.endeward@chemie.uni-frankfurt.de (B.E.); prisner@chemie.uni-frankfurt.de (T.F.P.); 3N.N. Vorozhtsov Institute of Organic Chemistry SB RAS, 630090 Novosibirsk, Russia; polienko@nioch.nsc.ru

**Keywords:** self-assembled complexes of nucleic acids, EPR spectroscopy, DEER/PELDOR, distance measurements, molecular dynamics

## Abstract

Pulsed electron–electron double resonance (PELDOR) spectroscopy is a powerful method for determining nucleic acid (NA) structure and conformational dynamics. PELDOR with molecular dynamics (MD) simulations opens up unique possibilities for defining the conformational ensembles of flexible, three-dimensional, self-assembled complexes of NA. Understanding the diversity and structure of these complexes is vital for uncovering matrix and regulative biological processes in the human body and artificially influencing them for therapeutic purposes. To explore the reliability of PELDOR and MD simulations, we site-specifically attached nitroxide spin labels to oligonucleotides, which form self-assembled complexes between NA chains and exhibit significant conformational flexibility. The DNA complexes assembled from a pair of oligonucleotides with different linker sizes showed excellent agreement between the distance distributions obtained from PELDOR and calculated from MD simulations, both for the mean inter-spin distance and the distance distribution width. These results prove that PELDOR with MD simulations has significant potential for studying the structure and dynamics of conformational flexible complexes of NA.

## 1. Introduction

Nucleic acids (NAs) play a vital role in biological events involving the storage of genetic information and transmission into proteins. NAs can fold into various secondary structures, which allow for the fine regulation of numerous processes in the body. Furthermore, self-assemblies of NAs can happen through the classical mechanism and non-canonic interaction between nucleobases, displaying several conformations and topologies with unique properties [1,2,3,4,5,6,7,8]. Such structures are found in genomic DNA, small RNA, and free-floating RNAs and affect catalytic processes, riboswitches, gene regulation, etc. [1,3,4,5,6,8,9,10,11,12]. The ability of NA to form supramolecular assemblies opens up new frontiers in NA-based artificial systems for diagnostics, therapy, nanorobotics, sensor development, and fundamental biological process investigations. For example, DNA-based nanoconstructions, especially DNA origami technology and aptamers, have been useful in the recognition of biological targets, drug delivery, bioimaging, etc. [13,14,15,16]. Many approaches are used to construct three-dimensional NA-based nanostructures with desired size, geometry, shape, and surface groups [13,17,18,19,20,21,22]. Therefore, NA structure, dynamics, and conformational distribution may give insight into its functions and interactions with ligands, proteins, and cell systems. Information on biological and synthetic DNA structures, conformational dynamics, and interactions is a powerful tool for the programmed self-assembly of NA, leading NA-based nanopharmaceuticals and biomimetics production. However, the study of NA self-assembly is challenging as spontaneous complexation makes the interpretation of conformations and topologies more complex [4,19,23]. Structural characterization of NA requires methods with sufficient resolution to catch changes.

Pulsed electron–electron double resonance (PELDOR) or double electron–electron resonance (DEER) spectroscopy allow for nanometer-distance measurements between paramagnetic tags attached to the specific sites of biomolecules [11,24,25,26,27,28]. PELDOR may be used for highly dynamic systems, providing distance distributions connected with conformational flexibility, which cannot be obtained by traditional methods [9,29,30]. PELDOR is suitable for NA transformation studies by measuring distances between spin labels at the 1.5–8 nm range [11,31]. Long-distance measurement was shown to range up to 10–16 nm for perdeuterated protein and RNA, which is a complicated procedure [9,32]. Angular, structural, and dynamic information can be estimated from these experiments. The combination of a powerful technique such as PELDOR with molecular modeling opens up unique possibilities for understanding the three-dimensional structure and dynamics of NA complexes.

Nitroxyl radicals are used for PELDOR studies for complexes of NAs due to their small size, easy chemical variations of functional group design for further bioconjugation, and high stability in vitro [11,33,34]. For the spin labeling of NAs, we have used the commonly known phosphorothioate oligonucleotide modification [35], which was introduced at any desired inter-nucleotide phosphate (Figure 1A). It opens up the possibility of attaching the nitroxide in any place of the oligonucleotide chain [31,36,37]. Furthermore, spin labeling with nitroxide without an extended linker results in narrow size distributions (Figure 1A) [31,36,37]. To demonstrate the feasibility of this approach, using the PELDOR method, we studied nitroxide-labeled oligonucleotide with phosphorothioate modification, which forms a telomeric G-quadruplex oligonucleotide complex [31]. Previously, we analyzed the formation of self-limited complexes of oligonucleotides by gel electrophoresis, atomic force microscopy, and molecular dynamics (MD) simulations and found a variety of three-dimensional structures that were formed by a pair of oligonucleotides [23].

Here, we used the PELDOR method and MD simulations to study the formation of nitroxide-labeled, three-dimensional, self-assembled complexes of oligodeoxyribonucleotides. Several oligonucleotides with flexible linker groups of varying lengths T_n_ (n = 0, 1, 3, 7, 10, 25) were utilized to demonstrate the feasibility of investigating such dynamical systems (Figure 1B). Unlike complexes with short linkers, oligonucleotides with long-sized linkers (n = 10 and 25) may “breathe”, forming either a straight or a loop-like structure and changing the distances between its primary chains (Figure 1D). The dynamics of complexes were estimated by size distribution in the PELDOR experiment, which is in excellent agreement with the results of MD simulations. Furthermore, the decrease in the EPR signal modulation depth of the samples with increasing linker length can give insights into the flexibility of complexes of NAs, leading to a loss of interaction between nitroxide tags. The results prove that the combination of PELDOR with MD has great potential for the study of highly polymorphic structures of NAs.

## 2. Materials and Methods

### 2.1. Materials

2,5-Dihydro-3-iodomethyl-2,2,5,5-tetramethyl-1*H*-pyrrol-1-oxyl was synthesized according to the literature procedure [38]. Sodium perchlorate, 99% (Acros Organics, MA, USA, cat. no. 197120010), 2-(N-Morpholino)-ethanesulfonic acid hydrate (MES, 99.5%, Sigma-Aldrich, Waltham, MA, USA, cat. no. 85H5735), and acetonitrile HPLC grade (≥99.93%, Sigma-Aldrich) were used. All solvents and other reagents, unless stated differently, were purchased from Sigma at the highest available grade and used without purification. Milli-Q water was used for the procedures. All oligonucleotides were purchased from the synthetic biology laboratory ICBFM SB RAS, Novosibirsk, Russia. Oligonucleotides with phosphorothioate modification were synthesized using standard solid-phase chemical synthesis and purified by RP-HPLC using a standard protocol.

### 2.2. Oligonucleotide Spin Labeling

The spin-labeled oligonucleotide was synthesized using nitroxyl radical derivative according to the procedure previously published by our group [31], adapted from Qin P.Z. et al. [36,37]. Briefly, 105 µL of 0.24 mM oligonucleotide with phosphorothioate modification, 15 µL of 1 M MES buffer solution (pH 5.8), and 30 µL of 0.1 M acetonitrile solution of 2,5-dihydro-3-iodomethyl-2,2,5,5-tetramethyl-1*H*-pyrrol-1-oxyl were mixed under stirring (800 rpm). The reaction mixture was incubated in the dark (wrapped with aluminum foil) at 25 °C for 24 h under stirring (600 rpm). Afterward, the reaction mixture in tubes was evaporated in vacuo in a SpeedVac centrifuge for 35 min for acetonitrile evaporation. Oligonucleotides were precipitated with 2% LiClO_4_ in acetone, washed using pure acetone, and desiccated under a vacuum. The oligonucleotides were dissolved in water and stored at −20 °C. The concentration of oligonucleotides was determined by UV spectroscopy on a UV-1500 spectrometer (Shimadzu, Kyoto, Japan). The oligonucleotide concentration was estimated using its absorbance at 260 nm and extinction coefficient ε = 220,300 M^−1^ cm^−1^ for M-T_3_* (5′-CTAACTAACG**TT**[PS]**T**CCATCATATG-3′) and ε = 227,100 M^−1^ cm^−1^ for N-T_3_* (5′-CGTTAGTTAG**TT**[PS]**T**CATATGATGG-3′). The oligonucleotide yield was 90%. The labeling efficiency of the oligonucleotide was estimated by CW EPR (par. 2.6) and was determined to be ~93%.

### 2.3. Gel Electrophoresis

Oligonucleotides were analyzed by polyacrylamide gel electrophoresis using 15% polyacrylamide gel (acrylamide/N,N′-methylenebisacrylamide ratio 39:1) in tris(hydroxymethyl) aminomethane–borate buffer (89 mM tris and 89 mM boric acid, pH 8.3) and 15 mM magnesium acetate at a voltage of 17 V/cm during thermostating in a water bath at 5 °C (LKB Bromma 2219 Multitemp II, Bromma, Sweden). A double-stranded DNA ladder of 50–1000 bp (SibEnzyme, Novosibirsk, Russia) was used to determine the mobility of complexes. Stains-all dye (Sigma-Aldrich, Waltha, MO, USA) was used for gel staining.

### 2.4. Oligonucleotide Complexes Melting Analysis

Thermal denaturation experiments were carried out in 0.2 cm quartz cells using a Cary 300 Bio spectrophotometer (Varian, Melbourne, Australia). Melting curves were registered at the wavelengths 260, 270, and 330 nm in the 5–95 °C range. The absorbance at 330 nm was chosen as a baseline. The temperature was changed at a 0.5 °C/min rate. The maximum of the melting curve derivative was calculated as a melting temperature (T_m_). A 10 µM equimolar mixture of oligonucleotides with M and N sequences in 10 mM sodium cacodylate (pH 7.2), 100 mM NaCl, and 10 mM MgCl_2_ was used for the thermal denaturation analysis.

### 2.5. EPR Sample Preparation

To form an oligonucleotide complex, a spin-labeled oligonucleotide (M-T_3_* or N-T_3_*) was mixed with non-spin-labeled oligonucleotides (M-T_n_ or N-T_n_) at a ratio of 1:1 according to Table 1. Each oligonucleotide was prepared at a 25 µM concentration in deuterated PBS buffer (10 mM phosphate buffer, pH 7.4, 100 mM NaCl, 10 mM MgCl_2_) containing 20% V:V ethylene glycol-d_6_ as a glass-forming agent. After adding cryoprotectant, 10 μL of the samples were transferred into 1.6 mm outer-diameter quartz EPR tubes (Suprasil, Wilmad LabGlass, Vineland, NJ, USA) and flash-frozen in liquid nitrogen.

### 2.6. CW EPR

A sample for continuous-wave (CW) X-band EPR spectroscopy for spin concentration determination was prepared in a 20 µL BLAUBRAND^®^ intraMark micropipette (BRAND, Wertheim, Germany). The X-band CW EPR (9.9 GHz) spectra were recorded at room temperature on a Bruker E580 spectrometer (Bruker, Karlsruhe, Germany). Experimental settings were as follows: microwave power 0.9464 mW, time constant 81.92 ms, conversion time 81.92 ms, modulation frequency 100 kHz, modulation amplitude 0.1 mT, number of scans 5.

### 2.7. PELDOR Measurements

PELDOR experiments were performed at Q-band (33.8 GHz) and 50 K on a Bruker Elexsys E580 spectrometer equipped with an EN 5107D2 resonator, a 150 W TWT microwave amplifier (187 Ka, Applied Systems Engineering Inc., Fort Worth, TX, USA), an Oxford CF935 cryostat and an Oxford ITC503 temperature controller (Oxford Instruments, Oxford, UK). PELDOR experiments were performed using the standard four-pulse sequence with pulse lengths of 32 ns for the probe pulses (π/2 and π) and 12 ns for the pump pulse (π) [39,40]. The pump position was applied at the maximum of the EPR spectrum. The pump frequency was set 40 MHz below the cavity maximum (center frequency), and the detection was set 40 MHz above the cavity center frequency. All obtained data were processed by Tikhonov regularization and the DEERNet network using DeerAnalysis2022 [41,42].

### 2.8. Molecular Dynamics

Molecular dynamics (MD) simulations for oligonucleotide complexes were similar to those previously described by our research group [23]. Briefly, oligonucleotide complexes were constructed as a B-form DNA double helix and manually changed through UCSF Chimera software v. 1.15 [43]. The nitroxide spin label was generated using XLEaP software (AmberTools 17 software package) [44] with subsequent geometry optimization by the Hartree–Fock method based on quantum mechanical calculation in the 6-31G++ basis in Gaussian 09 software. The atoms’ charges were calculated by the RESP method [45]. AMBER force field parmbsc1 [46] and gaff were used for DNA and nitroxide, respectively. The missing force field parameters for the nitroxide label were generated using the parmchk module of AmberTools20. The MD simulations were performed in the explicit water shell (TIP3P model) in a cuboid box (12 Å from each atom of protein) using ionsjc_tip3p parameters [47] for ions via the pmemd.CUDA module of AMBER20 [44]. The oligonucleotide complexes were neutralized by sodium ions. An equilibrium trajectory of 100 ns was obtained and then analyzed using the cpptraj tool of AmberTools20 [44]. A hierarchical cluster analysis was used to determine the most represented structure in the trajectory. Molecular visualization was performed via UCSF Chimera software.

## 3. Results and Discussion

### 3.1. Oligonucleotide Synthesis

The sequences of DNA oligonucleotides are displayed in Figure 1B. The native and phosphorothioate-modified oligonucleotides were prepared via solid-phase synthesis and purified by reverse-phase high-performance liquid chromatography (RP-HPLC) using a standard protocol [48]. The oligonucleotides with the different sequence chains M and N have two complementary blocks, which form a double-stranded DNA (Figure 2). 

The oligonucleotides sequence was previously designed, and sequences were selected to avoid the formation of intramolecular hairpins or self-complementary complexes and efficiently form complexes of various structures [23]. The nucleotide linkers with varying length T_n_ (n = 0, 1, 3, 7, 10, and 25) between DNA blocks were selected. The self-assembled DNA complexes that formed from the different pairs of oligonucleotides with M and N sequences (Table 1) have different conformation, topology, and flexibility (Figure 1C,D) [23]. Nitroxide was attached to the phosphorothioate modification according to the previously published procedure using 2,5-dihydro-3-iodomethyl-2,2,5,5-tetramethyl-1*H*-pyrrol-1-oxyl (Figure 1A) [31]. The spin labeling efficiency was estimated from CW EPR (Appendix A) and was determined to be ~0.93 ± 0.05 spins per oligonucleotide. We introduced the spin label at the center of the linker T_3_ to minimize the influence of the nitroxide on complementary interactions between chains of oligonucleotides (Figure 1B–D). To compare the difference between the nitroxide label position in the chains M or N, we synthesized M-T_3_* and N-T_3_* oligonucleotides (Figure 1B) and performed additional T_3_*_T_0_, T_3_*_T_3_, and T_3_*_T_25_ experiments (Table 1). 

### 3.2. Electrophoretic and Thermal Denaturation Analysis of Self-Assembled Complexes of Oligonucleotides

Previous research by our group has established that M-T_n_ and N-T_n_ pairs of oligonucleotides can form dimer, tetra-, hexa-, and higher-order complexes, which highly depend on the linker size [23]. To form an oligonucleotide complex, a spin-labeled oligonucleotide was mixed with a non-spin-labeled oligonucleotide at a ratio of 1:1, according to Table 1. Before examining the oligonucleotide assembly using a PELDOR analysis, we confirmed the presence in the sample of the required complexes. The mixtures were analyzed by gel electrophoresis and thermal denaturation experiments (Appendix A). The high-mobility bands in the electrophoresis image (~30–100 base pairs (bp)) are single-stranded oligonucleotides or dimers (Appendix A). The presence of well-pronounced bands with average mobility (~200 bp) is associated with tetramers [23]. The low-mobility bands (>300 bp), in most cases, have low intensity and are related to higher-order complexes. Moreover, the large complexes are very flexible and thus make it possible for the presence of several conformations, leading to bands broadening (Appendix A, e.g., T_25__T_3_*). The type of complexes (dimer, tetramer, etc.) is in good correlation with previously published data (Table 1) [23].

Measuring the thermodynamic stability of complexes was performed by a thermal denaturation analysis, known as a melting curve analysis. The samples were heated through a 5–95 °C range of temperatures, while UV absorbance at 260, 270, and 330 nm was collected. Typical melting curves and their first derivative are presented in Appendix A. A melting temperature (T_m_), defined as the maximum of a melting curve derivative (Appendix A), was calculated for all studied mixtures (Table 1). The complex stability slightly increased as the linker size increased, which resulted in a rise in T_m_ from 38 °C to 43 °C for T_1__T_3_* and T_25__T_3_*, respectively. Interestingly, T_0__T_3_* and T_3_*_T_0_ samples had a high T_m_ of 43 °C due to the slightly different structure of the complex, provided by the presence of additional stacking interactions between the base pairs at the junction of two T_3_* oligonucleotides (Figure 1C). Complexes with one or two nitroxide labels (T_3__T_3_*, T_3_*_T_3_, and T_3_*_T_3_*) indicated the same T_m_ of 39 °C. In the sample without nitroxides (T_3__T_3_), a T_m_ of 38 ± 1 °C was obtained, which corresponds to no significant influence of the spin label into complexation. These results are in line with those obtained in the gel electrophoresis experiment. The similar bands and their intensity can be seen in the electrophoresis image for T_3__T_3_*, T_3_*_T_3_, and T_3_*_T_3_* samples (Appendix A). The complexes with different flexibility and nitroxide–nitroxide distance were specially selected to estimate the PELDOR method’s capabilities. The typical structures are dimers and tetramers for NA complexes synthesized in this work (Table 1). Unlike tetramers (Figure 1), the dimer cannot be visualized by the PELDOR method due to only one spin label. A low amount of higher-order complexes (e.g., hexamers) was obtained (Appendix A, Table 1). Hexamers have high distances between labels and cannot be detected by the PELDOR.

### 3.3. Distance Measurements by PELDOR

PELDOR experiments were performed for the oligonucleotide complexes in Table 1 to obtain information regarding the nitroxide–nitroxide distance and the distance distribution. Initially, we obtained intramolecular distances in T_n__T_3_* samples with n = 0, 1, and 3 (Figure 3). To compare the possible differences in nitroxide labels in the opposite oligonucleotide chain, T_3_*_T_n_ complexes were analyzed (Figure 3). The distance distributions from PELDOR signals were calculated using Tikhonov regularization [49] and using the DEERNet network [50]. In most cases, we found only slight differences between both methods (cf. Figure 3 and Appendix A). However, the Tikhonov regularization provides more reliable data in our case. Following this analysis, a maximum distance probability of 7.5 nm was found for the T_0__T_3_* sample, which matches well with the value of 7.6 nm obtained for the T_3_*_T_0_ sample (Figure 3).

Because the conversion from the time domain PELDOR time trace to the distance distribution is an ill-posed mathematical problem, there are some uncertainties in the obtained distance distribution function P(R) shown in the DEERNet results as shaded regions (Appendix A). Therefore, the minor distance peaks of 3–5 nm are comparable with the experimental uncertainty (cf. Figure 3 and Appendix A). Another important finding is that the width of the distance peak broadening at 7.6 nm exceeds the width expected from the rotamers of the covalently attached spin label. Because of an expected large mean distance of ~7.5 nm and a broad distance distribution, a long dipolar evolution time window of 9 µs and a low spin concentration were chosen to reduce contributions from the intermolecular background functions. With this time window, the mean distance could be reliably extracted [51], whereas the exact width and shape of the distance distribution are already beyond the means of an accurate determination. The absence of a linker in the nitroxide tag ensures low conformational flexibility of the label itself, leading to precise distance determination by PELDOR [31]. Thus, the distance distribution width can be related to the flexibility and the distribution between several conformations of complexes.

The PELDOR data obtained for the T_1__T_3_* sample displayed a broad distribution, with an intensive peak at 7.3 nm (Figure 3). T_1__T_3_* showed smeared traces in the gel electrophoresis image (Appendix A). It corresponds to several conformations of tetramer complexes that lead to a broad distance distribution. It is not surprising that the peak at 7.3 nm has a large width again, ranging from 5.5 to 8.5 nm. Similar results were obtained for the T_3__T_3_* and T_3_*_T_3_ samples (Figure 3). We performed the PELDOR experiment for the T_3_*_T_3_* sample to gain additional insights. For T_3_*_T_3_*, a short distance of 4.3 nm between two labels was obtained (Figure 3). Comparing the two results for T_3__T_3_* and T_3_*_T_3_*, it can be seen that the distance distribution of the T_3_*_T_3_* PELDOR signal is narrower, with a peak at about 4 nm and a width of ~1.5 nm.

The information gained from PELDOR can be used to assess the percentage of spin pairs relative to the total amount of spins in the sample. The number of spin pairs can be deduced from the signal drop from the beginning to the end of the time trace after removal of the intramolecular signal decay, called modulation depth (Δ). DeerAnalysis software was used to obtain the intermolecular background decay and modulation depth (Appendix A, background traces and modulation depth by intercept of this curve with y-axis). The modulation depth for the different constructs is summarized in Appendix A. A modulation depth of ~0.4 was observed for T_0__T_3_* and T_3_*_T_0_, corresponding to the expected modulation depth for two coupled spins. A systematic decrease in the modulation depth for samples with increasing linker length was observed (Appendix A). The experimental modulation depth values of T_7__T_3_* and T_10__T_3_* were 0.092 and 0.093, four times lower than for T_0__T_3_*. These differences could be attributed to a reduced formation of tetramers and the presence of dimers with a single spin tag or other complexes with high distances between nitroxides (Appendix A, Table 1). This lower modulation depth for the T_25__T_3_* and T_3_*_T_25_ samples is striking. Despite a significant amount of tetramers being observed through electrophoresis (Appendix A), the modulation depths for these two samples were only 0.068 and 0.047. These lower modulation depths might result from a broad and unstructured distance distribution. In this case, it is difficult to accurately determine the intermolecular background decay, which might result in an incorrect modulation depth. Modulation depth calculations have essential information for NA complexes conformation and stability studies along with distance distribution. The small modulation depths and noisier PELDOR signals (Appendix A) for the samples with the longer linker (T_7_, T_10_, and T_25_) result in less reliable distance distributions and a loss of structural information. Therefore, distance distributions P(R) should be taken qualitatively only for these samples (Appendix A).

### 3.4. Molecular Dynamics Simulations

A series of 100 ns MD simulations in an explicit solvent shell were performed to optimally explore the conformational space of self-assembled complexes of oligonucleotides and generate the distance distribution between nitroxide labels. We built the MD library files for the phosphorothioate-modified thymidine labeled by nitroxide residue to perform MD simulations. A phosphorothioate linkage leads to a new chiral center at the P-atom [52]. Indeed, the separation of phosphorothioated oligonucleotide diastereomers with one modification is a difficult task. We did not separate diastereomers and performed an MD analysis using both S- and R-isomers (labeled as Sp and Rp). For the tetramer complex (Figure 2), two phosphorothioated oligonucleotides form four possible variants of tetramers’ stereoisomers (Sp-Sp, Sp-Rp, Rp-Sp, Rp-Rp). Instead of four isomers, MD calculations of the two most different isomers (Sp-Sp and Rp-Rp) were used. To characterize the structural stability of the complexes, we analyzed the root-mean-square deviation (RMSD), root-mean-square fluctuation (RMSF), and interspin distances along MD trajectories (Appendix A). The distance values for T_3_*_T_0_ and T_3_*_T_3_* samples have a deviation of less than 1 nm and plateaued after 20–30 ns. Further analysis showed that increasing linker length results in interspin distance fluctuations of 2–6 nm (Appendix A), and high RMSD changes along the trajectories indicate the enlarged conformational flexibility of oligonucleotide complexes. This is in agreement with RMSF per residue values for the entire complex that is below 6 Å for complexes with the T_0_ linker, up to 10 Å for complexes with the T_1_ or T_3_ linkers, and not higher than 15 Å for complexes with 7–25 nt linkers (Appendix A). At the same time, for the 10 bp duplex blocks in the complexes, per-residue RMSF values are typically below 1.5 Å and have higher values only for the terminal base pairs (RMSF < 6 Å), corresponding to fraying.

Comparing distance values averaged over Sp and Rp isomers, all samples exhibited a similar behavior of interspin distance values (Appendix A). The cluster analysis of the MD trajectories allows us to distinguish several conformations for each tetramer complex. The most represented structures in the trajectory, which shows the most dominant whole structure of complexes, are performed in Appendix A. Comparing T_25__T_3_* and T_3_*_T_25_ complexes with others, all other samples provide neat symmetrical structures with some perturbations in the linker site, which is in good correlation with interspin distance data.

To verify the interspin distances measured by the PELDOR analysis, the nitroxide–nitroxide distance distribution calculation was performed during the whole MD trajectory for each sample. The distance distributions by MD and PELDOR (Figure 4) analyses show excellent agreement for T_0__T_3_*, T_3_*_T_0_, and T_3_*_T_3_*. For T_3__T_3_*, T_3_*_T_3_ samples, the MD distance distribution covers the area of the most intense distribution obtained by PELDOR. Most striking was the substantial similarity of the maximum distribution values and whole distance distribution for the samples by MD and PELDOR. Moreover, we expected that Sp-Rp and Rp-Sp distances should be between the distances for the extreme structures as Sp-Sp and Rp-Rp, which improves data convergence. The most surprising aspect of the data for T_0__T_3_*, T_3_*_T_0_, T_3__T_3_*, T_3_*_T_3_, and T_3_*_T_3_* is the promising agreement with the nitroxide–nitroxide distance for the most represented structures in the MD trajectory (Table 2, Appendix A) and the maximum of the major peak of the PELDOR distance distribution. Nevertheless, these results require further verification through additional research. Further analysis shows that sample T_7__T_3_* indicates a low correlation between PELDOR and MD data (Appendix A). The data for T_7__T_3_* is not surprising as almost no tetramers were found according to gel electrophoresis (Appendix A). Moreover, MD distribution extends from 4 to 9.5 nm with a maximum of 7.5 and 8.5 nm for Rp-Rp and Sp-Sp isomers, respectively. In this way, Appendix A shows the problem when the PELDOR time trace of 9 µs is not long enough. The T_7__T_3_* just barely ‘covers’ the smallest distances from the MD calculations. For the T_10__T_3_*, T_25__T_3_*, and T_3_*_T_25_ samples, MD distributions were stated to range from ~7 nm to 10–15 nm (Appendix A). However, the time traces by PELDOR were only 9 µs, which is not long enough to cover 10 to 15 nm long distances [9]. Therefore, distances for T_10__T_3_*, T_25__T_3_*, and T_3_*_T_25_ samples are probably artifacts from the limited dipolar evolution time window and cannot be interpreted (Appendix A).

## 4. Conclusions

In summary, the tandem of PELDOR experiments and MD simulations has significant potential as a valuable tool for examining three-dimensional self-assembled complexes of NA. Conversely, our comparison shows that the presented methods have powerful abilities to quantitatively determine the structures and conformational dynamics of such flexible complexes. MD simulations enrich the interpretation of sparse PELDOR distance measurements, helping to answer biological and biophysical questions and providing an ensemble of conformations, the most representative structures, structural diversity, and some stability data for studying nucleic acid systems. The experimental results of the gel electrophoresis and melting analysis are in good agreement with PELDOR and MD data and reveal additional insights into the interpretation of oligonucleotide systems. The approach used has length limits, which originated on the one hand from the limits of PELDOR measurements (typically 2–10 Å) and on the other hand from the size of the molecular system in MD simulation (up to 10^6^ atoms including solvent molecules) for reliable simulation. We wonder if these findings could be helpful for the further investigation of natural systems with highly polymorphic structures, such as I-motifs, DNA hairpins, loops, G-quadruplexes, etc., and their conformational changes upon interactions with ligands, drugs, and changes in environmental factors [31]. The proposed tandem of the methods has the potential for studying three-dimensional structures such as DNA origami or other supramolecular DNA complexes. Moreover, the data can be applied for synthetic NA agents and complexes for various fields of medicine, diagnostics, and bionanotechnology.

## Figures and Tables

**Figure 1 biomolecules-14-00887-f001:**
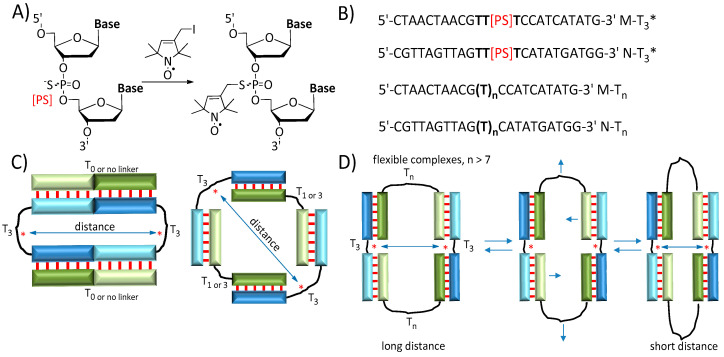
(**A**) Schematic representation of oligonucleotide spin labeling. [PS]—Phosphorothioate group. (**B**) Sequences of oligonucleotides. (**C**) Schematic representation of NA complexes with a short linker (T_0_, T_1_, T_3_) in one chain and a T_3_ linker with nitroxide. The spin labels are marked as a red star (*). (**D**) Schematic representation of flexible complexes with long linkers (T_7_, T_10_, T_25_).

**Figure 2 biomolecules-14-00887-f002:**
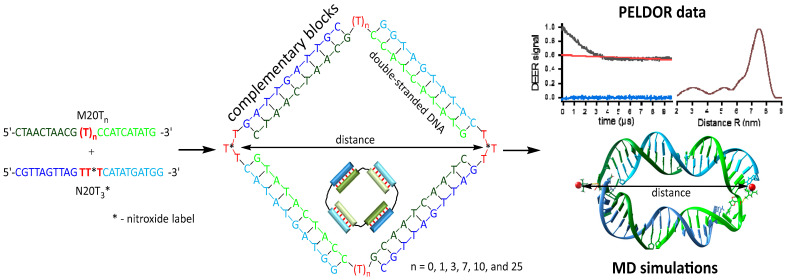
Schematic representation of a tetramer complex formed by a pair of oligonucleotides. Typical results of Q-band DEER measurements with the modulation depth cutoff, distance distribution profile, and three-dimensional structure as obtained by MD simulations.

**Figure 3 biomolecules-14-00887-f003:**
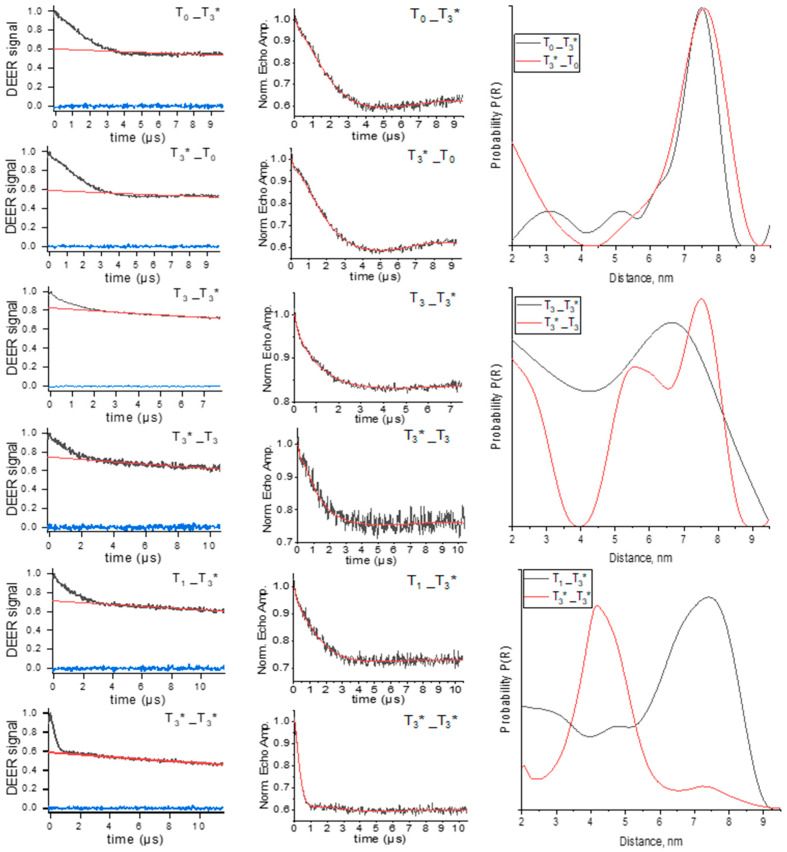
Q-band PELDOR results for T_0__T_3_*, T_3_*_T_0_, T_1__T_3_*, T_3__T_3_*, T_3_*_T_3_, and T_3_*_T_3_*. * means spin-labeled oligonucleotide. **Left**: PELDOR time traces (black) with the background fits (red) and an imaginary part of the signal (blue). **Middle**: Background-corrected PELDOR time trace (black) and Tikhonov fit (red). **Right**: The normalized distance distributions P(R) are obtained by Tikhonov regularization. For comparison, the data assessed by the DEERNet network using DeerAnalysis2022 software with uncertainty of distance are presented in Appendix A.

**Figure 4 biomolecules-14-00887-f004:**
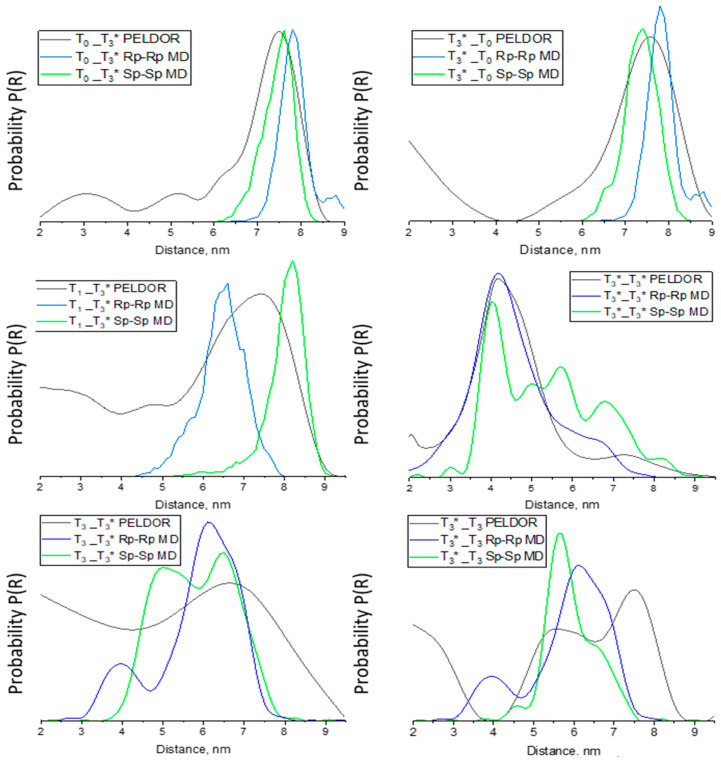
Spin–Spin distance distributions for T_0__T_3_*, T_3_*_T_0_, T_1__T_3_*, T_3__T_3_*, T_3_*_T_3_, and T_3_*_T_3_* from the Tikhonov regularization of PELDOR time traces and MD data analysis. * means spin-labeled oligonucleotide.

**Table 1 biomolecules-14-00887-t001:** Tested mixtures of oligonucleotides, type of complexes, and melting temperature.

Oligonucleotide ^1^		Abbreviation	Complex Type ^2^	Melting Temperature T_m_ (°C)
No 1	No 2			
M	N-T_3_*	T_0__T_3_*	t, h	43.0 ± 0.6
M-T_1_	N-T_3_*	T_1__T_3_*	t, (h)	38.5 ± 1.2
M-T_3_	N-T_3_*	T_3__T_3_*	d, t	39.2 ± 0.8
M-T_7_	N-T_3_*	T_7__T_3_*	d, (t)	41.0 ± 0.7
M-T_10_	N-T_3_*	T_10__T_3_*	d, (t)	42.5 ± 0.8
M-T_25_	N-T_3_*	T_25__T_3_*	d, t	43.0 ± 0.8
M-T_3_*	N-T_3_*	T_3_*_T_3_*	d, t	39.0 ± 1.0
M-T_3_*	N	T_3_*_T_0_	t, h	42.7 ± 0.6
M-T_3_*	N-T_3_	T_3_*_T_3_	d, t	39.0 ± 1.0
M-T_3_*	N-T_25_	T_3_*_T_25_	d, t	43.5 ± 0.8

^1^ The oligonucleotide sequences are 5′-CTAACTAACG(T)_n_CCATCATATG-3′ (M-Tn) and 5′-CGTTAGTTAG(T)_n_CATATGATGG-3′ (N-Tn); * means spin-labeled oligonucleotide. ^2^ Minor species in a sample are presented in brackets. d—dimer, t—tetramer, h—higher-order complexes (e.g., hexamer, etc.).

**Table 2 biomolecules-14-00887-t002:** The distances between nitroxides for the most representative structures in MD trajectories were obtained by cluster analysis for oligonucleotide complexes for Sp-Sp and Rp-Rp stereoisomers and the PELDOR distance peak maximum (Tikhonov regularization). * means spin-labeled oligonucleotide.

Sample	Distance MD, nm		PELDOR Distance, nm
	Sp-Sp	Rp-Rp	
T_0__T_3_*	7.2	8.0	7.5
T_3_*_T_0_	7.3	7.4	7.5
T_1__T_3_*	8.3	6.5	7.4
T_3__T_3_*	4.2	5.1	6.5–6.8
T_3_*_T_3_	5.9	6.4	5.6, 7.7
T_3_*_T_3_*	4.0, 5.3, 5.5, 5.7, 6.9, 7.4	3.4, 3.8, 3.9, 4.1, 5.0, 5.6	4.3
T_7__T_3_*	8.5	8.0	-
T_10__T_3_*	8.0	9.4	-
T_25__T_3_*	10.1	8.6	-
T_3_*_T_25_	13.3	13.0	-

## Data Availability

Data will be made available from the corresponding author upon reasonable request.

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
