# Peer review of "Pulsed Dipolar EPR for Self-Limited Complexes of Oligonucleotides Studies"

_biomolecules, 2024, doi:10.3390/biom14080887_

Round 1

Reviewer 1 Report

Comments and Suggestions for Authors

Authors presented a study entitled, Pulsed Dipolar EPR for Self-limited Complexes of Oligonucleotides Studies. In this work, authors integrated the experimental technique with theoretical in silico methods to gain an understanding of the conformational flexible complexes of Nucleic acids. The study investigated the formation of self-assembled complexes by including nitroxide spin labels to oligonucleotides and measuring the distance distributions obtained from PELDOR practically and calculating theoretically via MD simulations between chains to gain insights into conformational mobility. 

Following comments need to be addressed:-

1. In the introduction part, the authors should include the diagnostic and therapeutic Self-limited Complexes of Oligonucleotides in the current existence and their application. This can be mentioned with suitable examples.

2. PELDOR is an established technique to study the structural properties of complexed macromolecules and the challenge is spin-labeling. [PS] – phosphorothioate group. Did the authors studied P-NMR to support the stability of phosphate group tagged to the electrophilic groups?

3. In the molecular dynamics part, of course RMSD can give idea about the distances. Authors also mentioned that 'We haven’t separated diastereomers and performed MD analysis'. This is quite appreciable. However, while studying the conformational flexibilities, it is very important to determine, RMSF. Authors need to address this issue, seriously.

4. The experimental part is well supported by the data as reflected in manuscript and supporting information.

5. References are well organized and followed the guidelines

6. Results including figures and tables were presented clearly and discussed adequately.

7. The conclusion is in alignment with the aim of the study. 

Author Response

Thank you for the valuable suggestions and comments. We have carefully examined the comments and suggestions and revised the manuscript accordingly. We presented the Word file with track changes. Please find as follows the responses to the comments. Please note that all the comments are bold-faced, and the authors' reply follows immediately below the comments.

  1. In the introduction part, the authors should include the diagnostic and therapeutic Self-limited Complexes of Oligonucleotides in the current existence and their application. This can be mentioned with suitable examples.

Thank you for your valuable comment. We have inserted some examples with references in the Introduction section.

  1. PELDOR is an established technique to study the structural properties of complexed macromolecules and the challenge is spin-labeling. [PS] – phosphorothioate group. Did the authors studied P-NMR to support the stability of phosphate group tagged to the electrophilic groups?

Phosphorothioate modification of oligonucleotide is a well-established technology in PCR. Phosphorothioates are stable under various conditions. That is why we haven’t studied it by 31P NMR.

Spin labeling of phosphorothioate is a known procedure that was previously published. The spin-labeled oligonucleotide was synthesized using nitroxyl radical derivative according to previously published by our group procedure [1] adapted from Qin P.Z. et al. [2,3].

  1. Sannikova, N.E.; Kolokolov, M.I.; Khlynova, T.A.; Chubarov, A.S.; Polienko, Y.F.; Fedin, M.V.; Krumkacheva, O.A. Revealing Light-Induced Structural Shifts in G-Quadruplex-Porphyrin Complexes: A Pulsed Dipolar EPR Study. Phys. Chem. Chem. Phys. 2023, 25, 22455–22466, doi:10.1039/D3CP01775C.
  2. Qin, P.Z.; Haworth, I.S.; Cai, Q.; Kusnetzow, A.K.; Grant, G.P.; Price, E.A.; Sowa, G.Z.; Popova, A.; Herreros, B.; He, H. Measuring nanometer distances in nucleic acids using a sequence-independent nitroxide probe. Nat. Protoc. 2007, 2, 2354–2365, doi:10.1038/nprot.2007.308.
  3. Zhang, X.; Xu, C.X.; Di Felice, R.; Sponer, J.; Islam, B.; Stadlbauer, P.; Ding, Y.; Mao, L.; Mao, Z.W.; Qin, P.Z. Conformations of Human Telomeric G-Quadruplex Studied Using a Nucleotide-Independent Nitroxide Label. Biochemistry 2016, 55, 360–372, doi:10.1021/acs.biochem.5b01189.
  1. In the molecular dynamics part, of course RMSD can give idea about the distances. Authors also mentioned that 'We haven’t separated diastereomers and performed MD analysis'. This is quite appreciable. However, while studying the conformational flexibilities, it is very important to determine, RMSF. Authors need to address this issue, seriously.

Thank you for your valuable comment. We have analyzed the RMSF of the whole complexes and duplex blocks. We have inserted data in Supplementary materials and added discussion in section “3.4. Molecular Dynamics Simulations”. Complexes flexibility is in agreement with RMSF per residue values for the entire complex that is below 6 Å for complexes with the T0 linker, up to 10 Å for complexes with the T1 or T3 linkers, and not higher than 15 Å for complexes with 7–25 nt linkers (Figure 8). At the same time, for the 10 bp duplex blocks in the complexes, per-residue RMSF values are typically below 1.5 Å and have higher values only for the terminal base pairs (RMSF < 6 Å), corresponding to fraying.

  1. The experimental part is well supported by the data as reflected in manuscript and supporting information.
  2. References are well organized and followed the guidelines
  3. Results including figures and tables were presented clearly and discussed adequately.
  4. The conclusion is in alignment with the aim of the study. 

Thank you for the positive feedback.

Reviewer 2 Report

Comments and Suggestions for Authors

In this manuscript, authors combined PELDOR and MD simulation as a tandem tool for nucleic acid structure analysis. The results include experimental, and simulations well supported the statements authors made, there are several comments I have since this manuscript is aiming to develop a platform for DNA structure analysis.

I suggest authors consider these comments and try to address in discussion:  

1.     In this manuscript, how does authors define the DNA sequence? Does GC content affect the structural formation?

2.     For DNA structure complex, does this tandem method work for only 2-dimentional structures, or have the potential of 3D structures such as origami?  

3.     Is there a length limitation? Either upper and lower of the DNA sequence.

Author Response

Thank you for the valuable suggestions and comments. We have carefully examined the comments and suggestions and revised the manuscript accordingly. We presented the Word file with track changes. Please find as follows the responses to the comments. Please note that all the comments are bold-faced, and the authors' reply follows immediately below the comments.

  1. In this manuscript, how does authors define the DNA sequence? Does GC content affect the structural formation?

In this paper, we used previously characterized by other methods DNA/DNA complexes formed by similar native oligonucleotides. The oligonucleotide sequence was designed to avoid the formation of intramolecular hairpins or self-complementary complexes. We demonstrated duplex and linker lengths on the type and complex molecularity [1]. In general, the GC content of duplex blocks affects the hybridization efficiency of oligonucleotides. The variation of the linker block will influence its flexibility and thus complex type, size, and molecularity.

  1. Zamoskovtseva, A.A.; Golyshev, V.M.; Kizilova, V.A.; Shevelev, G.Y.; Pyshnyi, D.V.; Lomzov, A.A. Pairing nanoarchitectonics of oligodeoxyribonucleotides with complex diversity: concatemers and self-limited complexes. RSC Adv. 2022, 12, 6416–6431, doi:10.1039/d2ra00155a. 
  1. For DNA structure complex, does this tandem method work for only 2-dimentional structures, or have the potential of 3D structures such as origami?  

As you see from the Figure S8, the studied complexes are not two-dimensional structures. Duplexes form three-dimensional structures helices that do not lie in one plane and form a more complex three-dimensional geometry. That is why we can conclude that tandems of PELDOR and MD work for various three-dimensional nucleic acid structures. Thus, of course, it has high potential for studying 3D structures such as origami. We have added this point in the Conclusion section.

  1. Is there a length limitation? Either upper and lower of the DNA sequence.

As was mentioned in the Introduction section, PELDOR is suitable for NA transformation studies by measuring distances between spin labels in the 1.5-8 nm range [1,2]. The long-distance measurement was shown up to 10-16 nm on perdeuterated protein and RNA, which is a complicated procedure [3,4]. That is why, it only depends on what three-dimensional structures will be formed for the studied DNA sequence. If it is in an appropriate range, then it can be seen.

  1. Sannikova, N.E.; Kolokolov, M.I.; Khlynova, T.A.; Chubarov, A.S.; Polienko, Y.F.; Fedin, M.V.; Krumkacheva, O.A. Revealing Light-Induced Structural Shifts in G-Quadruplex-Porphyrin Complexes: A Pulsed Dipolar EPR Study. Phys. Chem. Chem. Phys. 2023, 25, 22455–22466, doi:10.1039/D3CP01775C.
  2. Collauto, A.; von Bülow, S.; Gophane, D.B.; Saha, S.; Stelzl, L.S.; Hummer, G.; Sigurdsson, S.T.; Prisner, T.F. Compaction of RNA Duplexes in the Cell**. Angew. Chemie - Int. Ed. 2020, 59, 23025–23029, doi:10.1002/anie.202009800.
  3. Endeward, B.; Hu, Y.; Bai, G.; Liu, G.; Prisner, T.F.; Fang, X. Long-range distance determination in fully deuterated RNA with pulsed EPR spectroscopy. Biophys. J. 2022, 121, 37–43, doi:10.1016/j.bpj.2021.12.007.
  4. Jeschke, G. DEER Distance Measurements on Proteins. Annu. Rev. Phys. Chem. 2012, 63, 419–446, doi:10.1146/annurev-physchem-032511-143716.

If we discuss the assembly of the complex, it is quite complex. It is affected by both the length of the complementary part, the length of the linker, etc. A variety of structures can be formed, as was shown earlier [1].

1.         Zamoskovtseva, A.A.; Golyshev, V.M.; Kizilova, V.A.; Shevelev, G.Y.; Pyshnyi, D.V.; Lomzov, A.A. Pairing nanoarchitectonics of oligodeoxyribonucleotides with complex diversity: concatemers and self-limited complexes. RSC Adv. 2022, 12, 6416–6431, doi:10.1039/d2ra00155a.

Reviewer 3 Report

Comments and Suggestions for Authors

Comments on the Quality of English Language

no comments, see attached file

Author Response

Thank you for the valuable suggestions and comments. We have carefully examined the comments and suggestions and revised the manuscript accordingly. We presented the Word file with track changes. Please find as follows the responses to the comments. Please note that all the comments are bold-faced, and the authors' reply follows immediately below the comments.

  1. Figure 1: For a better understanding, authors should give images of dimers and tetramers, as they gave on figure S2.

In our paper, we studied only the tetramers by PELDOR and MD. That is why in Figure 1 we presented different types of tetramers. It can’t be seen dimers and hexamers for such oligonucleotides by PELDOR due to the one spin-label in the complex and too high distances between nitroxides, respectively.

  1. Lines 89-90: I am not sure that the word “mobility” is pertinent here, when the authors comment on the modulation depth. This is actually related to discussion p7.

Thank you for your valuable comment. We have revised the text.

  1. Lines 54-55: “PELDOR may be used for highly dynamic systems, providing distance distributions connected with conformational flexibility and cannot obtained by traditional methods”: something is missing: “… and which cannot be obtained …”.

Thank you for your valuable comment. We have revised the text.

  1. Overall, PELDOR data do not show evident oscillations, although the time traces are unusually long. As a result, the distance distributions are quite large, and I feel that the authors do not mention that a lot …

We expected long distances between spin labels, so we used long-time traces. However, it is a standard procedure for the PELDOR method and may be found elsewhere.

  1. Figure 3: please make the legends/ticks bigger.
  2. Figure 3: the graphical figure with the modulation depth is not really useful.
  3. Figure 3: The T3*-T3 Tikhonov fit (red) does not exactly look like the one mentioned in figure S5. Maybe the authors made a mistake here by choosing the wrong fit. Please check.
  4. Figure 3: The T3*-T3* Tikhonov fit (red) looks like the DeerNet one of figure S5. Please check.

Thank you for your valuable comment. We have revised Figure 3. We used a bit more line width for Figure 3. That is why it may look not the same as in Figure S5 but the pictures are correct.

  1. Line 276: I don’t think the large distribution in the 2.5 to 4.5 nm is significant

Thank you for your valuable comment. We have revised the text.

  1. Lines 285-286: “However, when we investigate various biological systems, two spin-labeled oligonucleo-285 tides and their site-specific labeling may be complicated or impossible.”: Ido not understand this sentence, in particular in relation to what is written just before.

Thank you for your valuable comment. We have revised the text.

  1. Lines 338-340: “The distance distributions by MD and PELDOR (Figure 4), show excellent agreement for T0_T3*, T3_T0*, T3_T3*, T3*_T3, and T3*_T3*. Most striking was the substantial similarity of the maximum distribution values and whole distance distribution for the samples by MD and PELDOR.”: I disagree: For T3-T3* and T3*-T3, the MD and PELDOR data show basically very large distributions (see figure 4). I think the authors take out too much information compared to what one can say from the data.

Thank you for your valuable comment. We have revised the text.

  1. Table 2: T3*-T3* : MD simulations are not easily comprehensible by the figure in SI. Moreover, table 2 does not show the distribution of distances which go with the maximum value of the distribution. Does it make sense to give distances from MD at 3.04 ; 3.8 ; 3.9 ; 4.1 ? Maybe it would be better to give a range of distances ?

In Figure 4, we show the distance distribution, obtained by MD simulations, only between nitroxides. That is why this distribution may be compared with PELDOR data.

However, in Table 2, we provide data for the most representative structure by MD. This means that this structure was obtained by cluster analysis of MD. We have chosen the most representative structure (energy optimal structure) for all atoms in the molecule. After that, we calculated in this structure the distances between nitroxides. That is why distance values obtained from this structure may not be the same as previously. In this way, we cannot average these values as we have many different structures in MD simulation with different distances.

  1. No parameters are given to record the CW EPR spectra at room temperature. Please complete.

It was on par 2.2. However, we revised the text and added a new par 2.6. CW EPR.

  1. Could the authors explain why some curves in figure S4 are sometimes larger than others ? What is the wavelength used in this figure ?

We used the wavelength of 260 nm for UV melting analysis. Wavelength added in Figures S3 and S4 capture. The height and width of the first derivative of the UV melting curve depend on the type of complexes formed. In general, there is no correlation between the molecularity or geometry of self-limited complexes and the shape of the UV melting curve. In some cases, higher differential melting curves corresponded to the formation complex with cooperative interactions (stacking in the nicks for complexes without linker – T0) that enhance the thermal stability of the complex and make denaturation more cooperative (higher in magnitude and lower width). In the case of complexes with long linkers, an increase in linker length leads to the increased probability of terminal base pairs staking associated with loop formation by T3* linker.

Round 2

Reviewer 3 Report

Comments and Suggestions for Authors

Response to the comment 4:

I agree on the long time trace. But there are still no well defined oscillations even for long distances. I would have liked to read that type of comments in the paper.

Author Response

Thank you for the valuable suggestions and comments. We have carefully examined the comments and suggestions and revised the manuscript accordingly. We presented the Word file with track changes. Please find as follows the responses to the comments. Please note that all the comments are bold-faced, and the authors' reply follows immediately below the comments.

Response to the comment 4: I agree on the long time trace. But there are still no well defined oscillations even for long distances. I would have liked to read that type of comments in the paper.

Thank you for your valuable comment. We revised the text in section 3.3.